# Non-GABA sleep medications, suvorexant as risk factors for falls: Case-control and case-crossover study

Yoshiki Ishibashi[1,2☯*], Rie Nishitani[2☯*], Akiyoshi Shimura[3,4], Ayano Takeuchi[2], Mamoru Touko[2], Takashi Kato[5], Sahoko Chiba[2], Keiko Ashidate[2], Nobuo Ishiwata[2], Tomoyasu Ichijo[2], Masataka Sasabe[2]

1 Department of Internal Medicine, Kudanzaka Hospital, Company Overview of Federation of National Public Service Personnel Mutual Aid Associations, Chiyoda-ku, Tokyo, Japan, 2 Department of Preventive Medicine and Public Health, School of Medicine, Keio University, Shinjuku-ku, Tokyo, Japan, 3 Department of Psychiatry, Tokyo Medical University, Shinjuku-ku, Tokyo, Japan, 4 Department of Sleep and Psychiatry, Kanno Hospital, Wako-shi, Saitama, Japan, 5 Department of Pharmacy, Kudanzaka Hospital, Company Overview of Federation of National Public Service Personnel Mutual Aid Associations, Chiyoda-ku, Tokyo, Japan

☯ These authors contributed equally to this work.
* ishiyoshi414@gmail.com (YI); rie.hatakeyama@gmail.com (RN)

**Data Availability Statement:** All relevant data are within the manuscript and its Supporting Information files.

## Abstract

The aim of this study was to examine the risk of falls associated with the use of non-gamma amino butyric acid (GABA) sleep medications, suvorexant and ramelteon. This case-control and case-crossover study was performed at the Kudanzaka Hospital, Chiyoda Ward, Tokyo. A total of 325 patients who had falls and 1295 controls matched by sex and age were included. The inclusion criteria for the case group were hospitalized patients who had their first fall and that for the control were patients who were hospitalized and did not have a fall, between January 2016 and November 2018. The internal sleep medications administered were classified as suvorexant, ramelteon, non-benzodiazepines, benzodiazepines, or kampo. In the case-control study, age, sex, clinical department, the fall down risk score, and hospitalized duration were adjusted in the logistic regression model. In the case-control study, multivariable logistic regression showed that the use of suvorexant (odds ratio [OR]: 2.61, 95% confidence interval [CI]: 1.29–5.28), nonbenzodiazepines (OR: 2.49, 95% CI: 1.73–3.59), and benzodiazepines (OR: 1.65, 95% CI: 1.16–2.34) was significantly associated with an increased OR of falls. However, the use of ramelteon (OR: 1.40, 95% CI: 0.60–3.16) and kampo (OR: 1.55, 95% CI: 0.75–3.19) was not significantly associated with an increased OR of falls. In the case-crossover study, the use of suvorexant (OR: 1.78, 95% CI: 1.05–3.00) and nonbenzodiazepines (OR: 1.63, 95% CI: 1.17–2.27) was significantly associated with an increased OR of falls. Similar patterns were observed in several sensitivity analyses. It was suggested that suvorexant increases the OR of falls. This result is robust in various analyses. This study showed that the risk of falls also exists for non-GABA sleep medication, suvorexant, and thus it is necessary to carefully prescribe hypnotic drugs under appropriate assessment.

**Funding:** The authors received no specific funding for this work.

**Competing interests:** Rie Nishitani, Ayano Takeuchi, Mamoru Touko, Takashi Kato, Satoshi Ohtani, Sahoko Chiba, Keiko Ashidate, Tomoyasu Ichijo, Takanori Shirai, Nobuo Ishiwata, Masataka Sasabe. declares that they have no conflicts of interest. Yoshiki Ishibashi, reports personal fees from Children and Future co., Ltd. Akiyoshi Shimura reports personal fees from Dainippon Sumitomo Pharma, Yoshitomiyakuhin, and Meiji Seika Pharma. This does not alter our adherence to PLOS ONE policies on sharing data and materials.

## Introduction

Falls are a major cause of fractures and accidents and are also associated with reduced activities of daily living (ADL) in elderly people [1–3]. It has been reported that over 70% of hip fractures are caused by falls [1] and some fractures occur in approximately 5–10% of all falls. Hip fractures occur in 1–2% cases of falls [2]. Furthermore, even if patients do not get injured, repeated falling is associated with reduced ADL [3]. When viewed from the perspective of hospital management, 70% of hospital accidents are caused by falls, and therefore prevention of falls is an important issue [4].

Oral intake of sleep medications is a major risk factor for falls. For example, a systematic review found that the use of four or more internal medicines was associated with about 1.7–2.7 times increased risk of falls in the elderly [5]. Particularly, sedatives or hypnotics are the most strongly linked to increased risk of falls among oral medications, and it is suggested that there is an increased risk of falls in internal medicine patients with one or more sedatives or hypnotics [5]. In the past, non-benzodiazepines were not considered to be associated with an increased risk of falls and fractures [6]. In recent years, however, several studies have shown that the risk of falls and fractures associated with non-benzodiazepines intake has increased similarly to that of benzodiazepines and it is apparent that the risk associated with non-benzodiazepines intake is higher compared to benzodiazepines [7, 8]. In a previous study, the prevalence of insomnia was 8–18% in the general population, and the prevalence of insomnia symptoms generally increased with age [9]. Many people with insomnia use sleep medications [10, 11]. There appears to be a trade-off here. Though the elderly are at a high risk of falling due to lack of focus of attention, muscle weakness, and sensory deficits [12], at the same time the prevalence of insomnia also increases [9, 11].

Since the mechanism of the orexin receptor antagonist is different from that of existing sleep medications, suvorexant has been studied as an option for insomnia [13–15]. In randomized control trials (RCTs), suvorexant has not been found to be associated with an increased risk of falls and fractures [16–19]. However, there are no studies that have investigated falls as a main outcome, and in previous research, fall cases were very few in the intervention group and the non-intervention group [15–19]. In addition, research on side effects of suvorexant, including falls and fractures are few [20], so the fact that only results of pre-approved clinical trials are available makes it difficult to estimate the cost-effectiveness of suvorexant [21]. Ramelteon, a melatonin receptor agonist, is increasingly being used in the United States and Japan and has become one of the options for treating insomnia [20]. Nonetheless, there are few reports investigating the risk of falls associated with the use of ramelteon [22]. A study that evaluated the standing balance after oral administration of ramelteon found no adverse effect on balance [23]. However, to our knowledge, there are no real clinical reports that have reviewed falls associated with ramelteon [24, 25]. Suvorexant and ramelteon are non-gamma amino butyric acid (GABA) receptor agonist sleep medications, so they are considered to have little muscle relaxant action and little risk for falls [24–26]. However, the half-life of suvorexant is longer than previous medications like zolpidem [14, 27], and some studies have showed that suvorexant and ramelteon have carry-over effects [14, 23]. Regardless, the risk of falling associated with suvorexant and ramelteon is unclear. Thus, the aim of this study was to examine the risk of falls associated with the use of non-GABA receptor agonist sleep medications. We hypothesized that the use of suvorexant and ramelteon would be associated with increased risk of fall.

## Methods

### Target population, study design, and strategy for matching

Retrospective case-control and case-crossover studies were conducted at the Kudanzaka Hospital (231 beds) in Chiyoda-ward, Tokyo. The inclusion criteria for the case group were

hospitalized patients who had their first fall between January 2016 and November 2018. Second or subsequent falls after hospitalization or outpatient falls were excluded. The clinical departments of Kudanzaka hospital were orthopedic surgery, internal medicine, general surgery, rehabilitation department, psychosomatic medicine, gynecology, urology, and dermatology. Inclusion criteria for the control group were patients who were hospitalized between January 2016 and November 2018 and had not fallen. Control group data were collected from 9,111 patients, and controls were randomly matched, based on age (plus or minus 3 years) and sex with a ratio of case:control = 1:4 (Fig 1). Source population in this study is the group who did not fall but was hospitalized in Kudanzaka hospital; so, we sampled the control from the group randomly [28]. In previous studies, matching was performed using gender and age [29, 30], and to reduce overmatching and bias, this study matched only with gender and age [31, 32]. We confirmed the robustness of the result in the case-crossover analysis to eliminate the risk that the matching is overmatching or limited and to reduce the bias of control selection [33].

## Data collection and definition in the case-control and case-crossover study

**Falls.** The fall of a patient was defined as the incident in which the body of a patient suddenly or unintentionally hits the ground or another location. Falls in hospitals were registered in incident reports submitted by nurses who found the fallen patients. Incident reports included the department, age of the patient, sex, date and time of the fall, and location. Incident reports were checked by the group leader, and two researchers confirmed the incident via checking the electronic medical records.

**Sleep medications.** The internal sleep medications administered were classified as suvorexant, ramelteon, non-benzodiazepines, benzodiazepines, or the oriental medicine (called

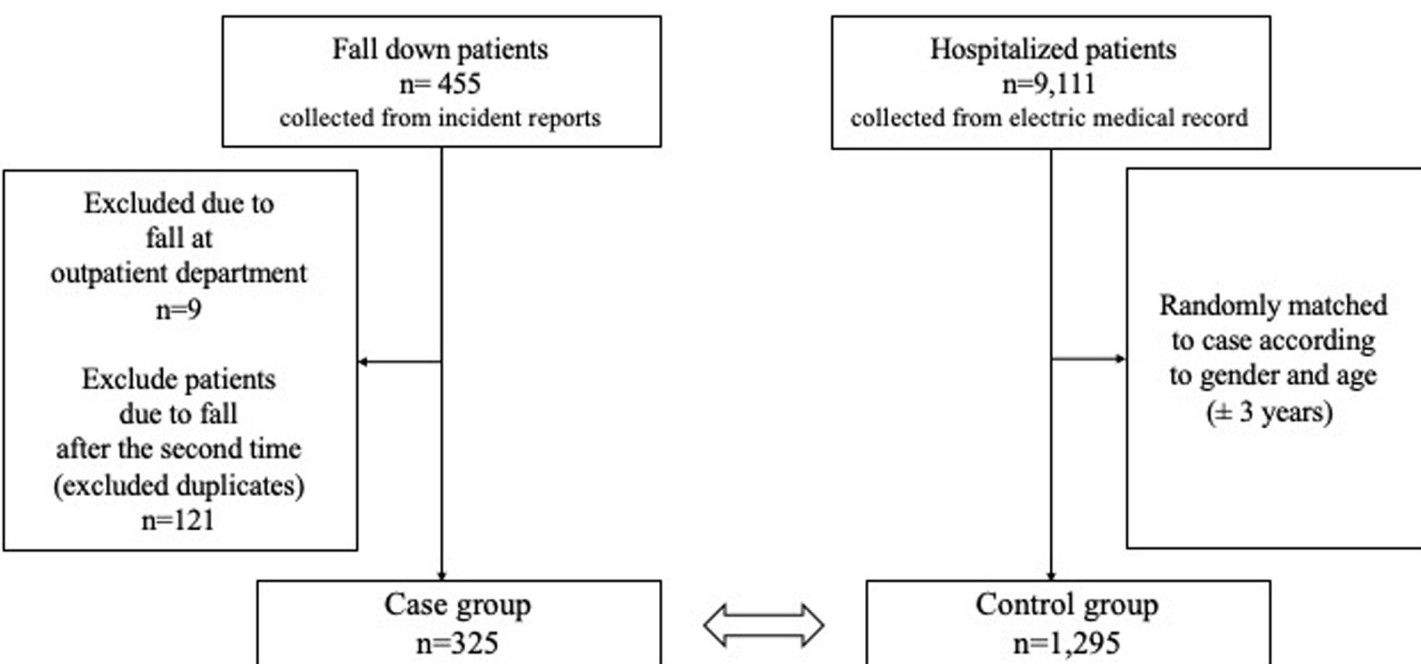

Falls in hospitals were registered in incident reports submitted by nurses who found the fallen patients. Incident reports include the department, age of the patient, sex, date and time of the fall, location, and the degree of injury. Control group data was collected from 9,111 patients hospitalized at the same term as the case group, and controls were randomly matched, based on age (plus or minus 3 years) and sex with a ratio of case: control = 1: 4.

**Fig 1. Design of case-control study.**

kampo, Yokukansan, or Yi-gan san). Kampo is a traditional and popular herbal drug in Japan and yokukansan is a kind of kampo [34]. Although, there is little medical evidence of its efficacy for insomnia, yokukansan is widely used in Japan [35, 36]. We analyzed these medications as exposure for falls. Adjustment for each medication was performed for patients taking multiple sleep medications, and the risk of fall was analyzed. The medications of the patients were extracted from the electronic medical record.

**Other variables.** The fall down risk score (it consists of 8 items namely age, medical history, vision or hearing impairment, motor dysfunction, activity, cognitive ability, medication, and excretion and was evaluated at 39 points as full score with 16 points or more considered as high risk of falls) [37, 38] (S1 Table). The fall down risk score was evaluated by nurses at hospitalized day and in every week. Hospitalized duration was defined as the date from hospitalization and discharge. The department of hospitalization, hospitalization duration, the fall down risk score, age, and sex were extracted from the electronic medical record.

**Variable measurement date in the case-control and case-crossover study.** In the case and control groups, the department was collected at hospitalized day, and hospitalization duration was recorded at discharge date. In the case group, we collected data of internal medicine measured on falling date, and the nearest fall down risk score before falls was collected. In the control group, since there were reports that falls tend to occur in approximately 1 week after hospitalization [39, 40], oral medications used within 1 week after hospitalization were investigated and the score of 1 week after hospitalization was collected. In the case-crossover study, internal medications of hospitalization and that of the middle date between hospitalization and falling were measured as an exposure of controls, and internal medications at falling was measured as an exposure of cases (S1 Fig). We summarized the variable information in the case-control and case-crossover study in the S2 Table.

## Ethical review

This study was approved by the Kudanzaka Hospital Ethics Committee. This research is part of a hospital-based quality improvement project, and data were dealt anonymously. The need for written informed consent from patients was waived because this study was conducted as part of a hospital-based quality improvement project and posed no risk to patients or their privacy.

## Statistical analysis

Sample size was calculated statistically, we needed approximately 1,000 patients at a significant level = 0.05, power = 0.9. For sensitivity analysis and case-crossover study, we collected data for more than 1,000 patients. Age, hospitalized duration, and the fall down risk score were analyzed as quantitative variables. Other data were analyzed as categorical variables. We conducted a complete-case analysis, so those with missing data were not included in the analysis. In the case-control study, age, sex, clinical department, the fall down risk score, and hospitalized duration were adjusted. In the case-crossover study, age, sex, clinical department, and the fall down risk score were adjusted. The association between sleep medications usage and falls was assessed by conditional logistic regression analysis using odds ratio (OR) and 95% confidence interval (CI).

In the case-control study, there were four analyses for sensitivity analysis. Firstly, since previous studies have shown a high risk of falls in orthopedic patients and those with musculoskeletal problems [5, 41, 42], an analysis excluding orthopedic surgery was conducted. Secondly, to eliminate the influence of multiple oral medications or long hospitalized duration, we excluded patients using multiple sleep medications or had long hospitalization. Thirdly,

evaluating the onset of action of benzodiazepines, analysis of benzodiazepines by separating them into long-acting type and short-acting type was carried out. Lastly, considering dose response, the analysis of the fall OR of non-benzodiazepines and benzodiazepines was conducted in the case-control study. We could not conduct the dose response analysis of suvorexant and ramelteon because the adoption of suvorexant in Kudanzaka Hospital was only 15 mg tablets and that of ramelteon was only 8 mg. All patients who were prescribed suvorexant and ramelteon were prescribed those of 15 mg and 8 mg. For multivariate analysis, the models considered the multicollinearity with the variance inflation factor of which the cutoff was 2.5. Statistical analysis was performed with R (http://www.r-project.org/) version 3.4.2. We conducted a two-sided test, and the analysis was at α level 0.05.

## Results

### Case-control and case-crossover study

There were 455 cases of falls at the Kudanzaka Hospital during the period of study (Fig 1). We excluded nine cases that occurred at outpatient service and 121 cases that occurred after the second time of falls, and 325 patients were assigned as cases of study. Of the 325 patients, 181 (55.7%) were taking sleep medications. Of the 1,295 patients who matched on age and sex, 416 (32.1%) were taking sleep medications and were selected as controls. Two patients, a 100-year-old male and a 102-year-old male, could not be matched to four controls (we chose two and one patients for controls corresponding to these cases).

Details of internal medicine and the number of patients who took sleep medications are shown in Table 1. Non-benzodiazepine medications included zolpidem, eszopiclone, and zopiclone. Benzodiazepine medications included triazolam, etizolam, brotizolam, rilmazafone, alprazolam, lorazepam, quazepam, flutoprazepam, clotiazepam, diazepam, flunitrazepam, nitrazepam, clonazepam, and lormetazepam. Across all sleep medications, the percentage of patients taking a medication was higher in the case group compared to the control group (p<0.001). Table 2 shows the measured background factors of the patients. Significant difference (p<0.001) was observed between the case group and the control group in terms of medical departments (except for general surgery and other departments), hospitalized duration, and the fall risk score. Therefore, the department of medical treatment, hospitalization duration, and the fall risk score were included in the multivariate analysis in the case-control study. Table 3 shows the result of conditional logistic regression analysis for falls. All variables were significant in univariate regression results. When the Bonferroni correction was performed in

**Table 1. Medication prescribed to cases and controls.**

| Medication | Cases n (%) | controls n (%) | P value |
|---|---|---|---|
| Suvorexant | 30 (9.2) | 20 (1.5) | <0.001 |
| Ramelteon | 13 (4.0) | 22 (1.7) | 0.017 |
| Non-benzodiazepines | 84 (25.8) | 151 (11.6) | <0.001 |
| Benzodiazepines | 96 (29.5) | 255 (19.7) | <0.001 |
| Kampo (yokukansan) | 21 (6.5) | 26 (2.0) | <0.001 |
| No sleep medication prescribed | 144 (44.3) | 879 (67.9) | <0.001 |
| Total patients | 325 | 1295 | |

Non-benzodiazepines include: Zolpidem, Eszopiclone, Zopiclone.

Benzodiazepines include: Triazolam, Etizolam, Brotizolam, Rilmazafone, Alprazolam, Lorazepam, Quazepam, Flutoprazepam, Clotiazepam, Diazepam, Flunitrazepam, Nitrazepam, Clonazepam, Lormetazepam.

Variables are analyzed by Fisher's exact probability test.

**Table 2. Patient background variables.**

| Characteristics | Case group | Control group | P value |
|---|---|---|---|
| Age, mean (±SD) | 76.7 (±12.0) | 76.1 (±11.7) | 0.391 |
| Sex n (%) (male) | 155 (47.7) | 615 (47.5%) | 0.951 |
| Department | | | |
| Orthopaedic surgery n (%) | 199 (61.2) | 649 (50.1) | <0.001 |
| Internal medicine n (%) | 59 (18.2) | 414 (32.0) | <0.001 |
| General surgery n (%) | 21 (6.5) | 123 (9.5) | 0.101 |
| Rehabilitation n (%) | 40 (12.3) | 62 (4.8) | <0.001 |
| Others(Gynecology, Urology, Dermatology, Psychology) n (%) | 6 (1.8) | 47 (3.6) | 0.118 |
| Hospitalized duration mean (SD) (day) | 70.1 (63.5) | 21.0 (27.9) | <0.001 |
| Fall down risk score mean (SD) (point) | 15.7 (4.18) | 11.3 (5.6) | <0.001 |

Continuous variables are analysed by student's t-test.

Percentage variables are analyzed by Fisher's exact probability test.

SD: standard deviation.

consideration of the multiple tests, almost the same result as the multivariate regression analysis was obtained. In the multivariate regression analysis, the use of suvorexant (OR: 2.61, 95% CI: 1.29–5.28), non-benzodiazepines (OR: 2.49, 95% CI: 1.73–3.59), and benzodiazepines (OR: 1.65, 95% CI: 1.16–2.34) was significantly associated with an increased OR of falls. However, the use of ramelteon (OR: 1.40, 95% CI: 0.60–3.16) and kampo (OR: 1.55, 95% CI: 0.75–3.19) was not significantly associated with an increased OR of falls. Table 4 shows the results of the case-crossover analysis, which showed that in the multivariate regression analysis, the use of suvorexant (OR: 1.78, 95% CI: 1.05–3.00) and non-benzodiazepines (OR: 1.63, 95% CI: 1.17–2.27) was significantly associated with an increased OR of falls. The use of ramelteon (OR:

**Table 3. Logistic regression model for sleep medications and covariances.**

| | Univariate analysis for fall OR (95%CI, P value) | Multivariate analysis for fall OR (95%CI, P value) |
|---|---|---|
| Sleep medications use | | |
| Suvorexant | 6.48 (3.66–11.7, <0.001) | 2.61 (1.29–5.29, 0.008) |
| Ramelteon | 2.41 (1.17–4.78, 0.013) | 1.40 (0.60–3.16, 0.429) |
| Non-benzodiazepines | 2.64 (1.95–3.56, <0.001) | 2.49 (1.73–3.59, <0.001) |
| Benzodiazepines | 1.71 (1.30–2.25, <0.001) | 1.65 (1.16–2.34, 0.005) |
| Kampo (yokukansan) | 3.37 (1.85–6.06, <0.001) | 1.55 (0.75–3.19, 0.233) |
| Department | | |
| Orthopaedic surgery (Reference) | | |
| Internal medicine | 0.46 (0.34–0.63, <0.001) | 1.19 (0.81–1.77, 0.367) |
| General surgery | 0.56 (0.33–0.89, 0.019) | 1.75 (0.93–3.17, 0.074) |
| Rehabilitation | 2.10 (1.36–3.21, <0.001) | 1.20 (0.69–2.06, 0.510) |
| Others(Gynecology, Urology, Dermatology, Psychology) | 0.42 (0.16–0.92, 0.047) | 1.42 (0.48–3.52, 0.481) |
| Hospitalized duration (per 1day) | 1.031 (1.027–1.035, <0.001) | 1.024 (1.020–1.029, <0.001) |
| Fall down risk score (per 1point) | 1.18 (1.15–1.21, <0.001) | 1.15 (1.11–1.19, <0.001) |

Multiple logistic regression adjusted age, sex, all sleep medications, department, hospitalized duration, fall down risk score.

OR: odds ratio, CI: confidence interval.

**Table 4. Case-crossover analysis.**

|  | Univariate analysis for fall OR (95%CI, P value) | Multivariate analysis for fall OR (95%CI, P value) |
|---|---|---|
| Suvorexant | 1.84 (1.10–3.07, 0.019) | 1.78 (1.05–3.00, 0.031) |
| Ramelteon | 1.14 (0.54–2.33, 0.707) | 1.19 (0.55–2.46, 0.640) |
| Non-benzodiazepines | 1.60 (1.16–2.20, 0.004) | 1.63 (1.17–2.27, 0.004) |
| Benzodiazepines | 1.13 (0.84–1.51, 0.419) | 1.09 (0.79–1.50, 0.585) |
| Kampo (yokukansan) | 1.55 (0.83–2.85, 0.159) | 1.46 (0.75–2.68, 0.277) |

Multiple logistic regression adjusted age, sex, hospitalized department, the fall down risk score and all sleep medications.

OR: odds ratio, CI: confidence interval.

1.19, 95% CI: 0.55–2.46), benzodiazepines (OR: 1.09, 95% CI: 0.79–1.50), and kampo (OR: 1.46, 95% CI: 0.75–2.68) was not significantly associated with an increased OR of falls.

## Sensitivity analysis

The results of the sensitivity analysis are shown in S3–S5 Tables. In the analysis excluding orthopedic surgery data, consistency of the results was maintained, and an increased OR of falls was observed for suvorexant (OR: 5.43, 95% CI: 1.59–20.1) and non-benzodiazepines (OR: 1.81, 95% CI: 0.92–3.45) (S3 Table). In order to eliminate the influence of multiple oral medications, patients using multiple sleep medications were excluded and there was also an increased OR of falls for suvorexant (OR: 2.48, 95% CI: 0.87–6.85), non-benzodiazepines (OR: 2.32, 95% CI: 1.44–3.70), and benzodiazepines (OR: 1.62, 95% CI: 1.05–2.47) intake (S4 Table). The analysis was performed excluding those whose length of hospital stay was above the median (S5 Table). An increase in the OR of falls was observed for suvorexant (OR: 9.09, 95% CI: 1.05–65.6) and non-benzodiazepines (OR: 2.34, 95% CI: 0.91–5.51). Ramelteon and Kampo could not be analyzed because the data were too sparse. However, the consistency of the results was maintained. The analysis in which benzodiazepines were divided into short-acting type and long-acting type is shown in S6 Table. Based on the adjusted analysis, there was an increased OR for falls for long-acting benzodiazepines (OR: 2.17, 95% CI: 1.30–3.57). Considering dose of medications, the consistency of the results was maintained (S7 Table). The multicollinearity test was performed, and the results are presented in S8 Table.

## Discussion

### Summary of results and differences in results from previous studies: Risk of falls with non-GABA medications

In the case-control study, the OR of falls was significantly higher in patients who used suvorexant, non-benzodiazepines, and benzodiazepines. The use of ramelteon and kampo was not associated with the significantly increased OR of falls. In the case-crossover study, the significance of benzodiazepines disappeared, but similar results were observed elsewhere. Benzodiazepines and non-benzodiazepines have a muscle relaxant action, for which the α1–3 subunits of the GABA receptor [43, 44] are responsible, and even in previous studies, an increased risk of falls has been shown [6–8].

Suvorexant is a sleep drug released in Japan and the United States in 2014 and has a new mechanism, blocking the orexin receptor that controls arousal [13, 15, 45]. Since it does not

act on GABA but acts selectively on the orexin system, it is considered not to have a muscle relaxant action or carry-over effect [26]. In some studies, its association with the risk of adverse events such as falls and fractures was negative [16–19]. However, in this study, an increase in the OR of falls was observed. In previous RCTs [17–19], it was difficult to statistically analyze for falls because the number of patients with falls was extremely small. The study that recorded most falls compared 12 (2.3%) and 8 (3.1%) falls [17], and it is necessary to collect falling patients and to design the case-control study for rare events such as falls. There is a possibility that risk determination is affected by the fact that the average age of the samples in this study is older (case group, mean age: 76.7 years), and the proportion of patients with orthopedic surgery is large. However, because the results did not change even in the analysis excluding orthopedic surgery, influence by the medical department is not anticipated. Analysis focused on elderly people has not been conducted in previous studies, and the risk of falls may be selectively high in elderly people. Clinical trials are performed for outpatients who have insomnia, but this study was performed in hospital settings and patients with comorbidities other than insomnia (no patients hospitalized for insomnia) were the subjects. For hospitalized patients, the risk of falls may be higher than that in outpatients. As far as we know, no research for suvorexant has been conducted on hard outcomes such as hip fracture, and further studies are needed. From the perspective of pharmacology, the reason falls are increased with suvorexant may be due to the half-life of the drug in the blood, which is about 9–13 hours [14]. Additionally, to maintain attention, the cholinergic system centered on the forebrain base plays an important role, and the orexin neural system interacts with it [46, 47]. Inhibition of the orexin receptor may reduce attention, and thus the orexin receptor antagonist, suvorexant, may increase the risk of falls by impairing attention [46]. In the elderly, orexin receptor and cholinergic neuron actions decrease, which may further increase the risk of falls [47]. S9 Table shows when patients fell in the case group. Patients who were prescribed suvorexant did not fall more at night than at daytime. There might be some residual sedation and carry over in the pharmacology of suvorexant [14].

Ramelteon is a melatonin receptor agonist, and it is increasingly being used by medical doctors [20, 22]. This case-control study did not show an increased OR for falls with the use of ramelteon. A similar result was obtained with the case-crossover analysis. Ramelteon might be less sedating, and previous studies have found less sedative adverse effects [23]; similar results were obtained in this study. Because melatonin stimulates daytime arousal and nighttime sleep [48, 49], it might not have raised the risk of falls. The result of this study, which was not significant, however, does not indicate the absence of risk [50]; therefore, attention must be paid to adverse effects such as falls. It is necessary to accumulate further research.

In insomnia guidelines, it is generally recommended to first consider cognitive-behavioral-therapy and psychological behavioral interventions and then consider hypnotic formulation [51, 52]. It is suggested that the risk of falls also exists for the non-GABA sleep medications as per the results in this study, and thus it is necessary to carefully prescribe hypnotic drugs under appropriate assessment.

## Evaluation of bias by sensitivity analysis and limitation of this research

In hospital studies, it is common that people with low severity are selected as controls, and the OR may be overestimated. Even in the case-crossover study, the same result as the case-control study indicates that bias by control selection is adjusted and the result is robust [33]. In the multivariate analysis, even if the fall risk score or hospitalized duration was added, the result did not change, and among those who are considered to have the same possibility of falls, the influence of sleep medications remained. Furthermore, conditional logistic regression analysis

excluding patients taking two or more kinds of sleep medications was similar, and it was considered that the effect of multiple sleep medications could also be eliminated.

There are several limitations in this research. First, we did not consider if patients were taking other medications, which have an increased risk of falls. This could be the bias of the result for this study. Second, it was impossible to evaluate the state of sleep and comorbidities in individuals. It cannot be denied that people with insomnia or chronic disease (like chronic obstructive pulmonary disease and obstructive sleep apnea) often take sleep medications and fall. However, most of the bias due to each patient's condition is controlled in a case-crossover study because the status of patients who have some chronic diseases is the same in the case-crossover comparison and the hospitalized duration. In future, a more planned prospective study is needed. We plan to conduct a validation study using instrumental variables to control potential confounders in this study. Third, this study might suffer from Berkson's bias due to recruitment from the hospitalized patients. Finally, the measurement and documentation of falls may not be entirely accurate and may be a limiting factor in this study.

In conclusion, sleep medications are associated with an increased OR of falls. This result is robust in various analyses, case-crossover studies, and sensitivity analyses. In this study, it was found that suvorexant was associated with an increased OR of falls. Although ramelteon was not associated with an increased OR of falls, further evaluation of its effect on the risk of falls and fractures in the future are needed. This study showed that the risk of falls also exists for non-GABA sleep medications, and thus it is necessary to carefully prescribe hypnotic drugs under appropriate assessment. The results of this study reaffirmed the importance of guidelines to optimize prescriptions. The preconception that there will be no falls because there is no muscle relaxation is dangerous. Physicians need to be cautious when prescribing sleep medications, even if they are not GABA receptor agonists.

## Supporting information

**S1 Fig. Design of case-crossover study.**
(PPTX)

**S1 Table. The fall down risk score.**
(XLSX)

**S2 Table. Summarized variable information in the case-control and case-crossover study.**
(XLSX)

**S3 Table. Sensitivity analysis of excluding orthopaedic surgery group.**
(XLSX)

**S4 Table. Logistic regression model among patients who were prescribed single medication.**
(XLSX)

**S5 Table. Sensitivity analysis of excluding longer hospitalized duration group.**
(XLSX)

**S6 Table. Logistic regression model for long and short benzodiazepine.**
(XLSX)

**S7 Table. Logistic regression model considering dose-response.**
(XLSX)

**S8 Table. Multicollinearity test in variables.**
(XLSX)

**S9 Table. Falling time according to sleep medications.**
(XLSX)

## Acknowledgments

The authors are grateful to Prof. T Takebayashi and Pro. T Okamura for helpful discussions. We also thank M. Abe and Ms. Miura for sharing their dataset with us.

## Author Contributions

**Conceptualization:** Yoshiki Ishibashi, Rie Nishitani, Akiyoshi Shimura.

**Data curation:** Yoshiki Ishibashi, Rie Nishitani, Mamoru Touko, Takashi Kato.

**Formal analysis:** Yoshiki Ishibashi, Ayano Takeuchi.

**Investigation:** Yoshiki Ishibashi, Rie Nishitani.

**Methodology:** Yoshiki Ishibashi, Rie Nishitani, Ayano Takeuchi.

**Project administration:** Yoshiki Ishibashi, Rie Nishitani.

**Resources:** Yoshiki Ishibashi.

**Software:** Yoshiki Ishibashi.

**Supervision:** Yoshiki Ishibashi, Akiyoshi Shimura, Ayano Takeuchi, Sahoko Chiba, Keiko Ashidate, Nobuo Ishiwata, Tomoyasu Ichijo, Masataka Sasabe.

**Validation:** Yoshiki Ishibashi.

**Visualization:** Yoshiki Ishibashi.

**Writing – original draft:** Yoshiki Ishibashi.

**Writing – review & editing:** Yoshiki Ishibashi, Rie Nishitani, Ayano Takeuchi, Sahoko Chiba, Keiko Ashidate, Nobuo Ishiwata, Tomoyasu Ichijo, Masataka Sasabe.

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
