## [Decision Letter · Decision Letter 0]

24 Jun 2020

PONE-D-20-05046

Non-GABA sleep medications suvorexant and ramelteon as risk factors for falls in elderly hospitalized patients: case-control and case-crossover study

PLOS ONE

Dear Dr. Ishibashi,

Thank you for submitting your manuscript to PLOS ONE. After careful consideration, we feel that it has merit but does not fully meet PLOS ONE’s publication criteria as it currently stands. Therefore, we invite you to submit a revised version of the manuscript that addresses the points raised during the review process.

ACADEMIC EDITOR: Please address all comments made by the reviewers.

We look forward to receiving your revised manuscript.

Kind regards,

Amir Radfar, MD,MPH,MSc,DHSc

Academic Editor

PLOS ONE

Journal Requirements:

"No: The funders had no role in study design, data collection and analysis, decision to publish, or preparation of the manuscript."

"The authors report no financial or other relationship that is relevant to this article. Rie Nishitani, Ayano Takeuchi, Mamoru Touko, Takashi Kato, Satoshi Ohtani, Sahoko Chiba, Keiko Ashidate, Tomoyasu Ichijo, Takanori Shirai, Nobuo Ishiwata, Masataka Sasabe. declares that they have no conflicts of interest. Yoshiki Ishibashi, reports personal fees from Children and Future co., Ltd. Akiyoshi Shimura reports personal fees from Dainippon Sumitomo Pharma, Yoshitomiyakuhin, and Meiji Seika Pharma."

4. Please include your tables as part of your main manuscript and remove the individual files. Please note that supplementary tables (should remain/ be uploaded) as separate "supporting information" files.

Reviewers' comments:

Reviewer's Responses to Questions

**Comments to the Author**

1. Is the manuscript technically sound, and do the data support the conclusions?

Reviewer #1: Yes

Reviewer #2: No

Reviewer #3: Yes

2. Has the statistical analysis been performed appropriately and rigorously? 

Reviewer #1: Yes

Reviewer #2: I Don't Know

Reviewer #3: No

3. Have the authors made all data underlying the findings in their manuscript fully available?

Reviewer #1: Yes

Reviewer #2: Yes

Reviewer #3: Yes

4. Is the manuscript presented in an intelligible fashion and written in standard English?

Reviewer #1: Yes

Reviewer #2: No

Reviewer #3: No

5. Review Comments to the Author

Reviewer #1: This is a timely and well designed and discussed study. There are a few questions that need to be elucidated

Most important of all is when were these falls? Were these middle of the night falls or falls the next day (due to residual sedation)? Second, ramelteon did not have increased risk of falls, but could that be because ramelteon is not sedating enough and not very effective as a sleep agent in the first place?

Reviewer #2: I read this manuscript with interest. The subject is important as so many people suffer from insomnia and they try multiple medications until they find something that works, and risk of fall is a serious side effect, especially in elderlies.

I would like to point out the following major concerns I have regarding this study design and writeup.

A) study design:

1.Data shows that the risk of side effects with both Z drugs and benzodiazepines is dose related. Higher doses tend to linger longer in the system and lead to fogginess, grogginess, altered postural states which have an increased likelihood of falls. Therefore, analysis without dose related side effects would affect the reliability and validity of the results and conclusion is questionable.

2. Authors did not mention in the limitation section if patients were taking other medications which have an increased risk of falls. This could again bias the results of the study.

3. Authors did not mention in the limitation section other confounders such as COPD and OSA (and how to control potential confounders) which could be exacerbated by benzodiazepines and affect the quality of sleep, hence increasing the risk of falls. Another thing to be taken in consideration is the hepatic and renal function of hospitalized patients which can prolong the drug metabolism, elimination, and action, enhancing the side effects on falls. Again, how were all these confounders controlled?

4. It is very possible that this study suffers from Berkson's bias due to recruitment from the hospitalized patients. This was not mentioned in the limitation section of the manuscript.

5. Inclusion and exclusion criteria are not clearly defined.

6. Pag 11 line 12: Regarding the statement “There were 46 people injured because of falls and this number was considered too small to analyze.”, were these patients with preexisting injury excluded from the study? Were these 46 patients on sleep medications?

7. Pag 14 lines 2-4: “Z” drugs were designed more selectively with the purpose of having less side effects than benzodiazepines. Authors assert that non-benzodiazepines cause more severe motor-impairment than benzodiazepines. This contradicts many other studies which report between 10 to 40 times increase in the myorelaxant effects in benzodiazepines compared to Z drugs. The statement in the paper needs to be revised.

B. Write up: I suggest having an English editing service review the manuscript:

8. Page 9 line 7: please correct “electric medical records”

9. Pag 11 line 2: What does the statement “An analysis of multiple oral patients was added to verify consistency of the results” mean?

10. Please check spelling for “case-crossover” throughout the manuscript. There are instances when the spelling is different.

11. The discussion section needs to be rewritten for cohesion and clarity of the conveyed information.

Reviewer #3: The concept of the research is clinically important, and the findings obtained in this research are also practical. However, the reviewer thinks that the description of methods was insufficient. There is also room for improvement in some statistical analysis designs.

6. PLOS authors have the option to publish the peer review history of their article (what does this mean?). If published, this will include your full peer review and any attached files.

Reviewer #1: Yes: Hrayr Attarian

Reviewer #2: No

Reviewer #3: No

---

## [Author Response · Author response to Decision Letter 0]

8 Aug 2020

August 7, 2020

Academic editor Amir Radfar

PLoS ONE

Dear Professor, Amir Radfar

Thank you for the opportunity to submit a revised version of our manuscript. Please see below our responses to the specific comments from the reviewers.

Response to Reviewers

#Reviewer 1

Most important of all is when were these falls? Were these middle of the night falls or falls the next day (due to residual sedation)?

Response: Thank you for your comments. We showed additional analysis of S Table 6. Patients who were prescribed long acting benzodiazepines fell more at night. Patients who were prescribed no sleep medication fell more at daytime. There was no significant difference of falls between daytime and night in patients who were prescribed other sleep medicines. We have modified the manuscript as per your advice as shown below.

S7 Table showed when patients fell in the case group. Patients who were prescribed suvorexant did not fall more at night than at daytime. There might be some residual sedation and carry over in the pharmacology of suvorexant [14]. (page 14 line10-12)

Second, ramelteon did not have increased risk of falls, but could that be because ramelteon is not sedating enough and not very effective as a sleep agent in the first place?

Response: Thank you for your comments. We agree with your proposal, so we have modified the manuscript as shown below.

Ramelteon might be less sedating, and previous studies have found less sedative adverse effects [23]; similar results were obtained in this study. (page14 line15-17)

#Reviewer 2

1. Data shows that the risk of side effects with both Z drugs and benzodiazepines is dose related. Higher doses tend to linger longer in the system and lead to fogginess, grogginess, altered postural states which have an increased likelihood of falls. Therefore, analysis without dose related side effects would affect the reliability and validity of the results and conclusion is questionable.

Response: Thank you for your comments. We added the analysis considering the dose of sleep medicine in the case-control study. S5 Table shows the results. The consistency of the results was maintained compared to Table 3. We have modified the manuscript as shown below.

Method

Lastly, considering dose response, the analysis of the fall OR of non-benzodiazepines and benzodiazepines was conducted in the case-control study. We could not conduct the dose response analysis of suvorexant and ramelteon because the adoption of suvorexant in Kudanzaka Hospital was only 15 mg tablets and that of ramelteon was only 8 mg. All patients who were prescribed suvorexant and ramelteon were prescribed those of 15 mg and 8 mg. (page10 line7-13)

Result

Considering dose of medications, the consistency of the results was maintained (S Table 5). (page12 line17-18)

2. Authors did not mention in the limitation section if patients were taking other medications which have an increased risk of falls. This could again bias the results of the study.

Response: Thank you for your comments. We have modified the manuscript according to your advice as shown below.

First, we did not consider if patients were taking other medications, which have an increased risk of falls. This could be the bias of the result for this study. (page 15 line14-16)

3. Authors did not mention in the limitation section other confounders such as COPD and OSA (and how to control potential confounders) which could be exacerbated by benzodiazepines and affect the quality of sleep, hence increasing the risk of falls. Another thing to be taken in consideration is the hepatic and renal function of hospitalized patients which can prolong the drug metabolism, elimination, and action, enhancing the side effects on falls. Again, how were all these confounders controlled?

Response: Thank you for your comments. We consider that most of the diseases for confounding were adjusted in the case-crossover study because the status of patients who have COPD, OSA, or some chronic disease is same in the case-crossover comparison. But it is unclear in the manuscript, we have modified it as shown below.

Second, it was impossible to evaluate the state of sleep and comorbidities in individuals. It cannot be denied that people with insomnia or chronic disease (like chronic obstructive pulmonary disease and obstructive sleep apnea) often take sleep medications and fall. However, most of the bias due to each patient's condition is controlled in a case-crossover study because the status of patients who have some chronic diseases is the same in the case-crossover comparison and the hospitalized duration. (page15 line16-21)

We totally agree with your suggestion that we should examine the hepatic and renal function of hospitalized patients. So, we conducted additional analysis in the case and control groups: Table A (below). The case group of albumin was worse compared to the control group. 

Table A Comparison of renal and hepatic function between cases and controls. 　

　 Cases Controls P value

Renal function 

　eGFR mL/min/1.73 m2 mean (SD) 69.2 (25.8) 65.3 (19.7) 0.013

　Creatinine mean (SD) 0.833 (0.427) 0.832 (0.353) 0.986

Hepatic function 

　Total bilirubin mean (SD) 0.752 (0.679) 0.814 (0.415) 0.123

　Platelet (×10000) mean (SD) 22.6 (8.7) 21.6 (7.1) 0.079

　PT% mean (SD) 94.9 (19.1) 95.5 (20.8) 0.679

　Albumin g/dl mean (SD) 3.69 (0.56) 3.93 (0.48) <0.001

Total patients 325 1295 　

We conducted additional sensitivity analysis including albumin in the case-control study (Table B). There was no change in the result compared to Table 3.

Table B Logistic regression model for sleep medications, covariances and albumin. 　

　 Univariate analysis for fall

OR (95%CI, P value) Multivariate analysis for fall

OR (95%CI, P value) 

Sleep medications use 

　Suvorexant 2.53 (1.26 - 5.12, 0.009) 

　Ramelteon 1.48 (0.63 - 3.39, 0.361)

　Non-benzodiazepines 2.53 (1.75 - 3.66, <0.001)

　Benzodiazepines 1.59 (1.11- 2.28, <0.001)

　Kampo (yokukansan) 1.43 (0.68 - 2.97, 0.340)

Department 

　Orthopedic surgery (Reference) 

　Internal medicine 1.14 (0.76 - 1.69, 0.528)

　General surgery 1.73 (0.86 - 3.30, 0.108)

　Rehabilitation 1.18 (0.67 - 2.03, 0.556)

　Others (Gynecology, Urology, Dermatology, Psychology) 1.59 (0.51 - 4.15, 0.376)

Hospitalized duration (per 1day) 1.023 (1.019 - 1.028, <0.001)

Fall down risk score (per 1point) 1.13 (1.09 - 1.18, <0.001)

Albumin (per g/dl) 0.40 (0.31 - 0.51, <0.001) 0.65 (0.48 - 0.88, <0.006)

Multiple logistic regression adjusted age, sex, all sleep medications, department, hospitalized duration, fall down risk score.

OR: odds ratio, CI: confidence interval.

4. It is very possible that this study suffers from Berkson's bias due to recruitment from the hospitalized patients. This was not mentioned in the limitation section of the manuscript.

Response: Thank you for your comments. We have modified the manuscript as per your advice as shown below.

Third, this study might suffer from Berkson's bias due to recruitment from the hospitalized patients. (page15-16 line22-1)

5. Inclusion and exclusion criteria are not clearly defined.

Response: Thank you for your comments. We have modified the manuscript as per your advice as shown below.

Abstract

The inclusion criteria for the case group were hospitalized patients who had their first fall and that for the control were patients who were hospitalized and did not have a fall, between January 2016 and November 2018. (page3 line5-7)

The inclusion criteria for the case group were hospitalized patients who had their first fall between January 2016 and November 2018. Second or subsequent falls after hospitalization or outpatient falls were excluded. (page7 line3-5)

Inclusion criteria for the control group were patients who were hospitalized between January 2016 and November 2018 and had not fallen. Control group data were collected from 9,111 patients, and controls were randomly matched, based on age (plus or minus 3 years) and sex with a ratio of case:control = 1:4 (Fig 1). Source population in this study is the group who did not fall but was hospitalized in Kudanzaka hospital; so, we sampled the control from the group randomly [28]. (page7 line7-12)

6. Pag 11 line 12: Regarding the statement “There were 46 people injured because of falls and this number was considered too small to analyze.”, were these patients with preexisting injury excluded from the study? Were these 46 patients on sleep medications?

Response: Thank you for your comments. The documents were unclear, so we deleted this description about injury. Injured falling cases were also included in the present analysis. We showed the data for injury and medication in this letter (Table C). This data included injured cases (46 cases) and matched controls (182 controls). According to the manuscript, one case, a 100-year-old male was matched with only 2 controls.

Table C Medication prescribed in injured cases and matched controls. 　

　 　 　 

Medication Cases n (%) Controls n (%) P value

Suvorexant 3 (6.5) 4 (2.2) 0.130 

Ramelteon 2 (4.3) 2 (2.8) 0.576

Non-benzodiazepines 14 (30.4) 16 (8.8) <0.001

Benzodiazepines 10 (21.7) 255 (20.1) 0.899

Kampo (yokukansan) 4 (8.7) 26 (2.2) 0.032

No sleep medication prescribed 23 (50.0) 124 (68.1) 0.022

Total patients 46 182 　

Non-benzodiazepines include: Zolpidem, Eszopiclone, Zopiclone.

Benzodiazepines include: Triazolam, Etizolam, Brotizolam, Rilmazafone, Alprazolam, Lorazepam, Quazepam, Flutoprazepam, Clotiazepam, Diazepam, Flunitrazepam, Nitrazepam, Clonazepam, Lormetazepam.

Variables are analyzed by Fisher's exact probability test.

7. Pag 14 lines 2-4: “Z” drugs were designed more selectively with the purpose of having less side effects than benzodiazepines. Authors assert that non-benzodiazepines cause more severe motor-impairment than benzodiazepines. This contradicts many other studies which report between 10 to 40 times increase in the myorelaxant effects in benzodiazepines compared to Z drugs. The statement in the paper needs to be revised.

Response: Thank you for your comments. We have modified the manuscript more simply as per your advice as shown below.

In the case-control study, the OR of falls was significantly higher in patients who used suvorexant, non-benzodiazepines, and benzodiazepines. The use of ramelteon and kampo was not associated with the significantly increased OR of falls. In the case-crossover study, the significance of benzodiazepines disappeared, but similar results were observed elsewhere. Benzodiazepines and non-benzodiazepines have a muscle relaxant action, for which the α1-3 subunits of the GABA receptor [43, 44] are responsible, and even in previous studies, an increased risk of falls has been shown [6-8]. (page13 line3-8)

B. Write up: I suggest having an English editing service review the manuscript:

Response: Thank you for your comments. We have modified the manuscript as per your advice.

8. Page 9 line 7: please correct “electric medical records”

Response: Thank you for your comments. We have modified the manuscript as per your advice.

from the electronic medical record.

9. Pag 11 line 2: What does the statement “An analysis of multiple oral patients was added to verify consistency of the results” mean?

Response: Thank you for your comments. We have modified the manuscript more clearly. It means we analyzed the patients excluding those who were prescribed multiple sleep medications. There was no description of the sensitivity analysis of hospitalized duration (S3 Table); we have also added the description.

Secondly, to eliminate the influence of multiple oral medications or long hospitalized duration, we excluded patients using multiple sleep medications or had long hospitalization. (page10 line3-5)

10. Please check spelling for “case-crossover” throughout the manuscript. There are instances when the spelling is different.

Response: Thank you for your comments. We have modified the manuscript as per your advice.

11. The discussion section needs to be rewritten for cohesion and clarity of the conveyed information.

Response: Thank you for your comments. We have modified the manuscript more simply and cohesive as per your advice.

Reviewer #3: The concept of the research is clinically important, and the findings obtained in this research are also practical. However, the reviewer thinks that the description of methods was insufficient. There is also room for improvement in some statistical analysis designs.

Response: Thank you for your comments. We have modified the manuscript totally as per your advice.

Response to the “Comments.”

Major comments

(1) The reviewer thinks that the title “Non-GABA sleep medications suvorexant and ramelteon as risk factors for falls in elderly hospitalized patients: case-control and case-crossover study” needs to be reconsidered because it gives the following misunderstandings and false impression.

Ramelteon increased the risk of falls…? (Although ramelteon did not significantly increase the risk of falls in any analyses.)

The study was intended to target the elderly from the beginning…? In that case, it is necessary to describe the range of target age in Methods.

Response: Thank you for your comments. We have modified the title as per your advice. Since we did not exclude the youth patients, we excluded the expression of “elderly.” We selected the expression that readers could understand that significant fall risk was observed only in patients who were prescribed suvorexant and not in those of ramelteon, as shown below.

“Non-GABA sleep medications, suvorexant as risk factors for falls: case-control and case-crossover study”

(2) The description of the Methods as a whole takes effort to understand. The reviewer thinks that the authors should separate the sections or paragraphs in a little more detail and clarify what is defined or explained. It may be an option to summarize the difference in the method of surveying items for each target group in a table.

Response: Thank you for your comments. We modified the method part totally as per your advice. We made a new S3 Figure summarizing the information of the variables and design.

Regarding statistical analysis, the reviewer thinks the authors should clarify which of the three statistical analyses is explained.

Response: Thank you for your comments. We have modified the statistical part more clearly according to your advice.

In Abstract, the description of Methods is also insufficient. The selection criteria of the target, the types of sleep medications investigated, the statistical analysis methods should be described.

Response: Thank you for your comments. We have added and clarified the method description in accordance with your advice.

(3) In the case-crossover analysis, age, sex and clinical department were adjusted, and 5 types of non-GABA sleep medications were included as independent variables. Since the case-control analysis showed that case-group had significantly higher fall risk scores, the reviewer thinks that the subscores (of 8 items) of the fall risk score (as factors that are likely to contribute to fall risk) should be included in the case-crossover analysis as independent variables.

Response: Thank you for your comments. We have modified the analysis adding the fall down risk score in the case-crossover study. We also modified Table 4 and the result and method description in the manuscript. The result did not change including the fall down risk score in the case-crossover study.

In addition, the reviewer thinks that the correlations between covariates should also be presented to evaluate confounding and to interpret the whole results.

Response: Thank you for your comments. We have added the analysis of the variance inflation factor (VIF) in the case-control study (S6 Table). There is no association in the exposure variables in this study. The variables are independent between each other. The cutoff of the VIF is 2.5 or 1.8 generally. *

*Johnston, R., Jones, K., & Manley, D. (2018). Confounding and collinearity in regression analysis: a cautionary tale and an alternative procedure, illustrated by studies of British voting behaviour. Quality & quantity, 52(4), 1957–1976.

We have added the description as shown below.

The multicollinearity test was performed, and the results are presented in S6 Table. (page12 line18-19)

Minor comments

(1) In Introduction, the authors mentioned attention decline as a cause of falls in the elderly. The reviewer thinks that muscle weakness and sensory deficits are also important factors.

Response: Thank you for your comments. We have added the description and changed the references [12], as shown below.

There appears to be a trade-off here. Though the elderly are at a high risk of falling due to lack of focus of attention, muscle weakness, and sensory deficits [12], at the same time the prevalence of insomnia also increases [9,11]. (page5 line20-22)

Reference

12. Institute of Medicine (US) Division of Health Promotion and Disease Prevention, Berg RL, Cassells JS, eds. The Second Fifty Years: Promoting Health and Preventing Disability. Washington (DC): National Academies Press (US); 1992.

(2) Yokukansan is sometimes written as Yi-gan san. It is better to add it?

Response: Thank you for your comments. We have added the description as shown below.

(called kampo, Yokukansan, or Yi-gan san) (page8 line5-6)

(3) The authors mentioned “the number of patients with falls was extremely small. The study which recorded most falls compared 12 and 18 falls” as limitations of previous studies (p. 14, line 16-17). What is the percentage?

Response: Thank you for your comments. We have added the description as shown below.

The study that recorded most falls compared 12 (2.3%) and 8 (3.1%) falls [17], and it is necessary to collect falling patients and to design the case-control study for rare events such as falls. (page13 line15-17)

(4) The authors hypothesized that falls due to the use of suvorexant might have induced REM sleep behavior disorder and led to falling from beds (p. 15, line 14-17). The reviewer wonders how many, in the authors’ data, fell from beds and what percentage of those were taking suvorexant.

Response: Thank you for your comments. Unfortunately, we have no data about falling from beds. So, we have deleted the description about beds, as shown below.

Additionally, to maintain attention, the cholinergic system centered on the forebrain base plays an important role, and the orexin neural system interacts with it [46, 47]. Inhibition of the orexin receptor may reduce attention, and thus the orexin receptor antagonist, suvorexant, may increase the risk of falls by impairing attention [46]. In the elderly, orexin receptor and cholinergic neuron actions decrease, which may further increase the risk of falls [47]. S7 Table shows when patients fell in the case group. Patients who were prescribed suvorexant did not fall more at night than at daytime. There might be some residual sedation and carry over in the pharmacology of suvorexant [14]. (page14 line5-12)

(5) Missing punctuation and capitalization were found. The reviewer also recommend the authors to proofread English by native speakers.

Response: Thank you for your comments. We have modified the manuscript totally. The new manuscript was checked by native speakers.

---

## [Decision Letter · Decision Letter 1]

18 Aug 2020

PONE-D-20-05046R1

Non-GABA sleep medications, suvorexant as risk factors for falls: case-control and case-crossover study

PLOS ONE

Dear Dr. Ishibashi,

Thank you for submitting your manuscript to PLOS ONE. After careful consideration, we feel that it has merit but does not fully meet PLOS ONE’s publication criteria as it currently stands. Therefore, we invite you to submit a revised version of the manuscript that addresses the points raised during the review process.

ACADEMIC EDITOR:Thank you for the revised version .Please address the comment made by reviewer # 3 for the final decision.

We look forward to receiving your revised manuscript.

Kind regards,

Amir Radfar, MD,MPH,MSc,DHSc

Academic Editor

PLOS ONE

Reviewers' comments:

Reviewer's Responses to Questions

**Comments to the Author**

1. If the authors have adequately addressed your comments raised in a previous round of review and you feel that this manuscript is now acceptable for publication, you may indicate that here to bypass the “Comments to the Author” section, enter your conflict of interest statement in the “Confidential to Editor” section, and submit your "Accept" recommendation.

Reviewer #1: All comments have been addressed

Reviewer #2: All comments have been addressed

Reviewer #3: All comments have been addressed

2. Is the manuscript technically sound, and do the data support the conclusions?

Reviewer #1: Yes

Reviewer #2: Yes

Reviewer #3: Yes

3. Has the statistical analysis been performed appropriately and rigorously? 

Reviewer #1: Yes

Reviewer #2: Yes

Reviewer #3: Yes

4. Have the authors made all data underlying the findings in their manuscript fully available?

Reviewer #1: Yes

Reviewer #2: Yes

Reviewer #3: Yes

5. Is the manuscript presented in an intelligible fashion and written in standard English?

Reviewer #1: Yes

Reviewer #2: Yes

Reviewer #3: Yes

6. Review Comments to the Author

Reviewer #1: (No Response)

Reviewer #2: (No Response)

Reviewer #3: All comments have been addressed and the manuscript has been properly revised.

Just one trivial point to be re-checked; S1 and S3 Figures are "Table", not "Figure".

7. PLOS authors have the option to publish the peer review history of their article (what does this mean?). If published, this will include your full peer review and any attached files.

Reviewer #1: **Yes: **Hrayr Attarian

Reviewer #2: No

Reviewer #3: No

---

## [Author Response · Author response to Decision Letter 1]

21 Aug 2020

August 19, 2020

Academic editor

Amir Radfar

PLoS ONE

Dear Professor, Amir Radfar

Thank you for the opportunity to submit a revised version of our manuscript. Please see below our responses to the specific comments from the reviewers.

Response to Reviewers

#Reviewer 3

Just one trivial point to be re-checked; S1 and S3 Figures are "Table", not "Figure".

Response: Thank you for your comments. We modified the manuscript, table, and figure as your suggestion.

---

## [Editor Report · Decision Letter 2]

24 Aug 2020

Non-GABA sleep medications, suvorexant as risk factors for falls: case-control and case-crossover study

PONE-D-20-05046R2

Dear Dr. Ishibashi,

We’re pleased to inform you that your manuscript has been judged scientifically suitable for publication and will be formally accepted for publication once it meets all outstanding technical requirements.

Kind regards,

Amir Radfar, MD,MPH,MSc,DHSc

Academic Editor

PLOS ONE
---

## [Editor Report · Acceptance letter]

28 Aug 2020

PONE-D-20-05046R2 

Non-GABA sleep medications, suvorexant as risk factors for falls: case-control and case-crossover study 

Dear Dr. Ishibashi:

I'm pleased to inform you that your manuscript has been deemed suitable for publication in PLOS ONE. Congratulations! Your manuscript is now with our production department. 

Kind regards, 

on behalf of

Dr. Amir Radfar 

Academic Editor

PLOS ONE